# A Blood-Based Molecular Clock for Biological Age Estimation

**DOI:** 10.3390/cells12010032

**Published:** 2022-12-21

**Authors:** Ersilia Paparazzo, Silvana Geracitano, Vincenzo Lagani, Denise Bartolomeo, Mirella Aurora Aceto, Patrizia D’Aquila, Luigi Citrigno, Dina Bellizzi, Giuseppe Passarino, Alberto Montesanto

**Affiliations:** 1Department of Biology, Ecology and Earth Sciences, University of Calabria, 87036 Rende, Italy; 2Biological and Environmental Sciences and Engineering Division (BESE), King Abdullah University of Science and Technology KAUST, Thuwal 23952, Saudi Arabia; 3Institute of Chemical Biology, Ilia State University, 0162 Tbilisi, Georgia; 4SDAIA-KAUST Center of Excellence in Data Science and Artificial Intelligence, Thuwal 23952, Saudi Arabia; 5National Research Council (CNR)—Institute for Biomedical Research and Innovation—(IRIB), 87050 Mangone, Italy

**Keywords:** sjTRECs, ELOVL2, age prediction, epigenetics

## Abstract

In the last decade, extensive efforts have been made to identify biomarkers of biological age. DNA methylation levels of ELOVL fatty acid elongase 2 (ELOVL2) and the signal joint T-cell receptor rearrangement excision circles (sjTRECs) represent the most promising candidates. Although these two non-redundant biomarkers echo important biological aspects of the ageing process in humans, a well-validated molecular clock exploiting these powerful candidates has not yet been formulated. The present study aimed to develop a more accurate molecular clock in a sample of 194 Italian individuals by re-analyzing the previously obtained EVOLV2 methylation data together with the amount of sjTRECs in the same blood samples. The proposed model showed a high prediction accuracy both in younger individuals with an error of about 2.5 years and in older subjects where a relatively low error was observed if compared with those reported in previously published studies. In conclusion, an easy, cost-effective and reliable model to measure the individual rate and the quality of aging in human population has been proposed. Further studies are required to validate the model and to extend its use in an applicative context.

## 1. Introduction

In the last decade, extensive efforts have been made to identify biomarkers of the biological ageing. They were mainly based on molecular methods and include the analysis of mitochondrial DNA 4977 deletion accumulations [1], telomere shortening [2], advanced glycation end products [3], and aspartic acid racemization [4]. All of these methods show several limitations due either to technical difficulties for the biomarker detection and/or to the moderate correlation between the assessed biomarker and the age of the sample under investigation. DNA methylation variants have been widely used as biomarkers of the rate of ageing and several mathematical models, the so-called “epigenetic clock”, have been developed to estimate the biological age [5,6,7,8]. By measuring the DNA methylation levels at some specific sites, they represent the most accurate measure of biological age and age-related disease risk available today. The biological age obtained from these epigenetic clocks has been found to be predictive of mortality [9,10,11,12,13] and other aging-related outcomes such as frailty [14] or cognitive and physical functioning [15,16,17,18,19]. Their main drawbacks include the tissue-specificity, and their still relatively high cost of the microarray technology on which they were based. More recently, DNA technology has triggered efforts toward the simplification of the array-based epigenetic clocks and several models have been developed to date. Among the markers included in these “simplified” clocks, ELOVL fatty acid elongase 2 (ELOVL2) represents a robust candidate gene. Several previous reports demonstrated that epigenetic variability of ELOVL2 regulatory region is highly correlated with age prediction [20,21,22,23,24]. In fact, within the top ten markers predictive of human epigenetic age, four are localized in the CpG islands in the regulatory element of the ELOVL2 gene, accounting for over 70% of the one of the most validated epigenetic clocks so far developed [25]. For a functional point of view, in vitro and in vivo studies demonstrated that epigenetic variability of ELOVL2 promoter contributes to the variability of the aging process by regulating lipid metabolism [26,27].

Another well-validated and important molecular marker of ageing is the signal joint T-cell receptor rearrangement excision circles (sjTRECs). sjTRECs are extra-chromosomal DNA by-products of the rearrangements of gene segments encoding the variable parts of TCR α and β chains during intra-thymic development. sjTREC is one particular TREC arising through an intermediate rearrangement in the TCRD/A locus in developing TCRαβ+ T lymphocytes [28]. Several lines of evidence showed an sjTREC decline in human peripheral blood with increasing age in several population samples [28,29,30,31], echoing the age-related thymic adipose involution in a life-long process and consequent thymic function loss. sjTRECs-based methods, exploiting the observed age-related decline of sjTRECs in human peripheral blood, show a relatively high prediction accuracy with the only limitation due to its tissue-specificity. A decrease in thymic output was also associated with increases in cancer incidence [32], infectious disease [33], autoimmune conditions [34], generalized inflammation [34], atherosclerosis [35] and all-cause mortality [36].

To the best of our knowledge, only one study attempted to build a molecular clock for biological age prediction exploiting both sjTRECs and ELOVL2 biomarkers [37]. The main drawback of this work was its reduced sample size that affects the generalizability and robustness of the reported findings.

The present study aimed to develop a more accurate molecular clock in a sample of the Italian population of different ages covering the whole span of adult life. In particular, we re-analyzed the EVOLV2 methylation data previously obtained in an Italian sample whose blood samples have been here analysed to quantify the amount of sjTRECs [38]. Then, based on the quantification of these two non-redundant biomarkers, a molecular model for biological age prediction has been proposed.

## 2. Materials and Methods

### 2.1. Samples

The dataset included 194 unrelated individuals (90 men and 104 women) with a mean age of 63.5 years. The recruitment of subjects older than 90 years was carried out between 2002 and 2005 through the population registries of Calabrian municipalities. Subjects in the age range 20–89 years were recruited between 2004 and 2007 as part of a survey aimed at monitoring the health status of this population segment in Calabria [39]. Fully written informed consent was obtained from all of the participants, and all studies were approved by the Ethics Committee of the University of Calabria.

### 2.2. DNA Samples Preparation

Peripheral blood samples were collected in EDTA containing tubes from each human subject. DNA was extracted from buffy coats following standard procedures. Genomic DNA was obtained by phenol/chloroform purification and then stored at −20 °C until use. DNA concentration and purity were determined spectrophotometrically using NanoDrop™ One (Thermofisher, Wilmington, DE, USA).

### 2.3. ELOVL2 Pyrosequencing Analysis

Methylation analysis of nine consecutive CpGs in the *ELOVL2* gene region was performed by bisulfite pyrosequencing. The detailed protocol also described by other authors was previously reported [38].

### 2.4. qPCR Assays

Real-time PCR conditions and primers for quantification of sjTREC were carried out according to the previously published technique [30]. In brief, Real-time PCR experiments were carried out using a QuantStudio 3 Real-Time PCR System (Applied Biosystems, Forster City, CA, USA) with a PowerUp SYBR Green Master mixture (Applied Biosystems, Forster City, CA, USA). To quantify sjTREC levels, a relative quantification method was used with the TATA box Binding Protein (TBP) housekeeping gene as an internal reference (dCt = CtTBP − CtsjTREC). Real-time PCR was performed on approximately 50 ng DNA in 25 μL reaction volumes, containing 500 nM of each primer. PCR conditions were: 95 °C for 30 s, then 95 °C for 5 s, 60 °C for 15 s, and 72 °C for 20 s, for 45 cycles. The dissociation curve analysis was performed using default setting temperature. Amplicon size was 140 bp for sjTREC and 113 bp for TBP. All reactions were performed in triplicate and the average value from each sample was used for further analyses.

### 2.5. Statistical Analysis

Pearson correlation coefficient and linear regression were used for assessing the univariate association between biological markers and age. 

We used JADBio (v1.3.32), an automated machine learning platform for creating machine learning models able to predict age from methylation markers [40]. JADBio methodology has been previously published [40]. In short, the platform performs an extensive search over several pre-processing, feature selection, and predictive modelling algorithms in order to find identify the best configuration for the task at hand. The hyper-parameters of each algorithm are also optimized during the search. Robust performance estimation protocols are used for estimating the predictive performance of the final model as well as for avoiding overfitting [41]. JADBio outputs include the best predictive model corresponding to the optimal configuration of algorithms and hyperparameters, the list of features selected for entering the model (signature), and an estimate of the predictive performance of the returned model. We applied JADBio twice, first including and then excluding sjTREC in the list of candidate predictors.

## 3. Results

Figure 1 and Table 1 show the age distribution of the sample under study.

### 3.1. Methylation of ELOVL2 Gene Promoter and Chronological Age

All of the analyzed CpG sites showed a strong positive correlation with age indicating that they could all be good estimators of the chronological age (Table 2).

From Table 2, it can be also observed that although all of the 9 CpG sites of ELOVL2 were significantly correlated with the age, the maximum correlation coefficient was detected for their average methylation value (r = 0.860, *p* < 0.001). For this reason, it will be used as an input predictor variable in the following developed models.

### 3.2. sjTREC Levels and Chronological Age

Figure 2 shows the correlation between sjTREC levels and age at the recruitment in the 194 analyzed blood samples.

The sjTREC content exhibits a significant age-related decline. In particular, a linear regression analysis between individual age and sjTREC level showed a strong negative dependence and that the formulated model explained a large and highly statistically significant proportion of the total age variance (R^2^ = 0.617). These prediction values seem consistent with those of the previous reports [28,30].

### 3.3. Development of a Blood-Based Molecular Clock

Finally, in order to obtain more accurate age prediction models, a machine learning approach was adopted. This approach produced two final best models, one using the average methylation value of the ELOVL2 promoter region as only predictor, and the coupling ELOVL2 with sjTREC. Both models were trained using Support Vector Regression Machines (SVR) with Polynomial Kernel as the best performing models. Their predictive performances are reported in Table 3. Further details on the two models are available on the JADBio platform.

Model with ELOVL2 and sjTREC:


https://app.jadbio.com/share/39e4c459-abd1-4765-887e-e93f18e4155d


Model with ELOVL2:


https://app.jadbio.com/share/8492f92b-39f7-48d7-99bc-79cbae478f37


The same analyses were repeated considering only males or females subjects. Overall, the results remained consistent, with a decrease in predictive performance for the sex specific models probably due to a decreased sample size. Particularly, the Mean Absolute Error (MAE) for the model containing both ELOVL2 and sjTREC passes from 4.449 years to 4.733 (only males) and 4.709 (only females). For the model with ELOVL2 solely, the reduction is from 4.954 years to 5.220 (males) and 5.055 (females).

Figure 3 shows the relationship between the actual and the predicted age by the model in which *ELOVL2* was used in combination with sjTREC. Predicted and chronological ages were highly correlated with a MAE of 4.440 (panel a). In line with other similar studies, older individuals (>60 years old) showed an increased difference between predicted and chronological age (residuals) compared to younger ones and their predicted age was rather underestimated (panel b) [38,42,43,44,45].

To assess the predictive capacities of the formulated models in the different stages of aging, the analyzed sample was grouped into six different age classes by intervals of 10 years (Table 4).

For both models, prediction accuracies decreased with advanced ages. However, the average methylation value of *ELOVL2* was used in combination with sjTREC, the SVR model exhibited much improved accuracy, particularly in the older age-groups. In fact, models with or without sjTREC showed similar prediction performances for individuals younger than 60 years, while in older individuals the SVR model also including sjTREC content showed an improved accuracy ranging from 6 months for the 60–70 age-group to about 1.3 years for individuals older than 80 years.

## 4. Discussion

We developed a machine learning model able to infer chronological age from the analysis of the methylation levels of *ELOVL2* and sjTREC quantification in blood samples of 194 individuals of different ages covering the whole span of adult life. We re-assessed the DNA methylation markers of *ELOLV2* gene together with the analysis of sjTREC content in the blood. We found that the sjTREC levels and DNA methylation of the *ELOVL2* gene are very useful markers for age prediction. In fact, an SVR model including these two non-redundant markers showed high prediction accuracy with a prediction error of about 4.4 years. Interestingly, this prediction error was extremely low in younger individuals with a MAE lower that 2.5 years and increased up to around 4.5 years in older subjects with the exception of very old individuals where the prediction error, in line with other literature data [46,47], resulted very high. Models with or without sjTREC showed similar accuracy for individuals with younger than 60 years, while in older individuals this accuracy significantly improves when the model also included sjTREC content (Table 4). The results here reported suggest that sjTREC in an independent predictor of chronological age and the usefulness of sjTREC as a supplementary marker for DNA methylation loci. Consequently, to obtain more accurate predictions, the inclusion of sjTREC is strongly required if these molecular models are applied to older subjects.

The model we proposed presents several points of strength. (i) It is based on two very accurate markers of the ageing process. In fact, ELOVL2 represents one of the most widely used markers for age prediction and does not show tissue-specificity, as observed for the most part of the epigenetic markers so far identified. sjTREC is a very reliable biomarker of thymic activity whose decline represents one of main hallmark of the immunosenescence process. (ii) The strategies adopted for the model development were based on strongly robust methods. In fact, in a recent comparative evaluation over 360 datasets, methods implemented in JADBio demonstrated to be able to avoid overfitting, with the cross-validated predictive performances estimated by the system being in line with out-of-sample estimation on separated datasets [40]. (iii) It represents a very cost-effective and portable method to measure biological age with the potential to extend its applicability not only in the forensic field, but also in the geriatric research where it might be used to identify possible interventions (pharmacological or nutritional) able to slow down the aging process. In fact, both markers can be analyzed using several available RT-PCR methods [30,48,49,50,51]. In addition, technology such as single base extension can be easily incorporated into well-established capillary electrophoresis systems to analyze methylation levels of the ELOVL2 gene [52,53,54]. As cost-effective approach, it might also allow for a re-analysis of the same samples, and the increase in the number of technical replicates (rarely performed in the published studies on this topic) has been demonstrated to clearly improve the detection of both markers and, consequently, the corresponding age prediction models [24]. While until some years ago most part of epigenetic clocks applied to measure the rate of ageing was based on the analysis of hundreds of CpG sites whose methylation levels were detected through microarray technologies, a recent work demonstrated that epigenetic models obtained analyzing only a small number of CpG sites provided highly comparable results with respect to epigenetic models estimated with array-based technologies [54]. In fact, their application in the geriatric clinical practice is constantly spreading [46,55]. (iv) Since sjTREC quantification does not require viable cells, it is well-suited for assessing thymic function in large population samples in which the collected blood samples are appropriately stored.

The limitations of our study deserve to be mentioned. The application of the proposed method is restricted to blood samples and body parts containing blood and is not possible for other body parts or fluids, such as semen or saliva, that do not contain T-cells in quantities required for sjTREC detection. Additional studies are therefore required to overcome this methodological limitation. Another point of weakness is the absence of information regarding longitudinal data that might be used for a further validation of the proposed model. However, since it is based on the joint use of the two markers whose individual accuracy to measure the rate of aging had already been largely demonstrated in previous studies, the impact on longitudinal outcomes such as mortality risk and loss of functional independence should be expected to be non-negligible.

In conclusion, the data here reported may be of interest, because an easy, cost-effective and reliable model was proposed that might represent a new tool to measure the individual rate and the quality of aging in geriatric ages. Further studies are therefore required to validate the model, particularly in an applied context.

## Figures and Tables

**Figure 1 cells-12-00032-f001:**
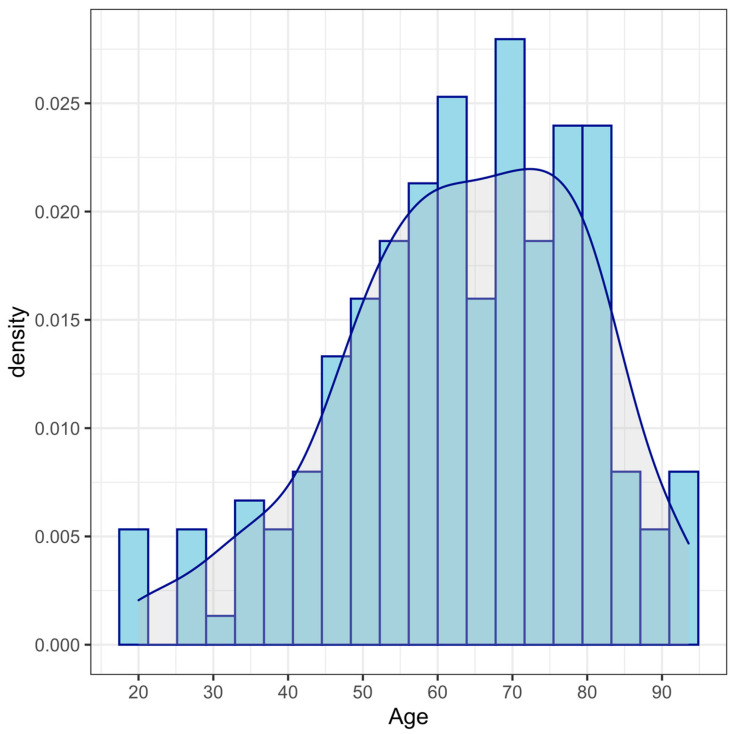
Frequency age distribution of the analyzed sample.

**Figure 2 cells-12-00032-f002:**
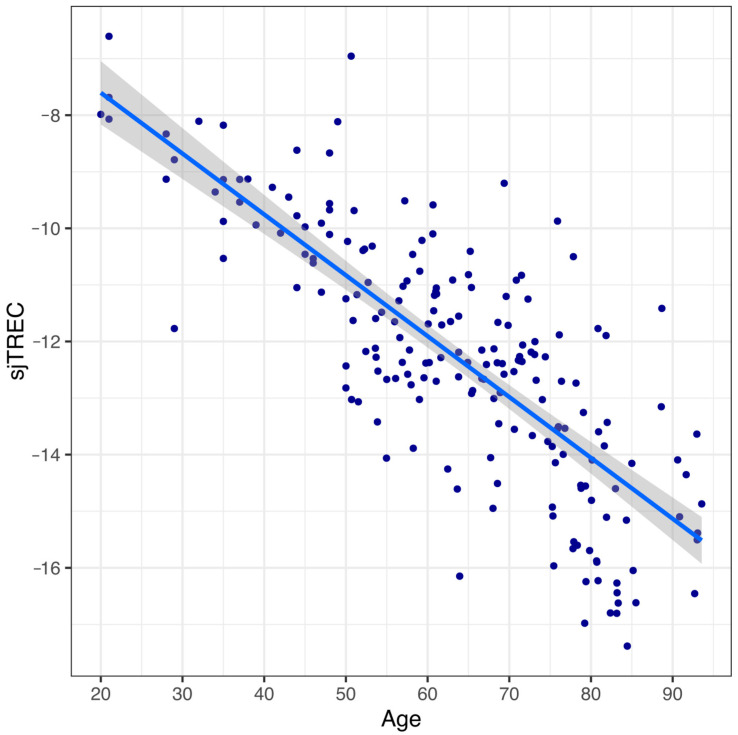
sjTREC levels in 194 blood samples of different ages (20–94 years old). sjTREC content declined progressively with age (R^2^ = 0.617).

**Figure 3 cells-12-00032-f003:**
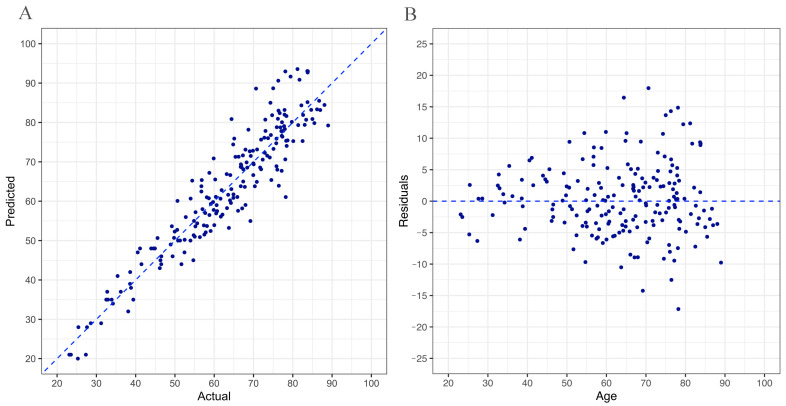
Age prediction from blood-derived DNA samples. (**A**) Linear regression of the relationships between human individual age and age predicted using the molecular clock in which ELOVL2 was used in combination with sjTREC (dotted lines correspond to 95% prediction interval). (**B**) Deviation of DNA methylation (predicted) age and chronological age (residuals) against the predicted age. The deviation in older individuals (>60 years old) was higher compared to younger ones. Linear and LOESS regression lines are reported in dashed red on the left and right panel, respectively.

**Table 1 cells-12-00032-t001:** Age distribution of the analyzed sample.

Cpg Site	Men (N = 90)	Women (N = 104)	Total Sample (N = 194)
Mean age (SD)	64.3 (15.7)	62.8 (16.9)	63.5 (16.3)
Median age	65.4	63.8	65.0
Age range	21.0–93.6	20.0–93.1	20.0–93.6

**Table 2 cells-12-00032-t002:** Correlation values between the 9 CpG sites of *ELOVL2*, their average value, and the age at the recruitment of the analyzed sample.

CpG Site	Chromosome Location (GRCh38)	r	*p*-Value
1	Chr6: 11,044,661	0.692	<0.001
2	Chr6: 11,044,655	0.758	<0.001
3	Chr6: 11,044,647	0.773	<0.001
4	Chr6: 11,044,644	0.811	<0.001
5	Chr6: 11,044,642	0.801	<0.001
6	Chr6: 11,044,640	0.793	<0.001
7	Chr6: 11,044,634	0.852	<0.001
8	Chr6: 11,044,631	0.751	<0.001
9	Chr6: 11,044,628	0.825	<0.001
Mean CpG value	Chr6: 11,044,661–11,044,628	0.860	<0.001

**Table 3 cells-12-00032-t003:** Prediction accuracies of the SVR models obtained in JADBIO using the average methylation value of the *ELOVL2* promoter region alone or in combination with sjTREC content.

Metric	Model without sjTREC	Model with sjTREC
	Mean Performance (95% CI)	Mean Performance (95% CI)
R-squared	0.805 (0.745, 0.844)	0.839 (0.753, 0.884)
Mean Absolute Error	4.954 (4.622, 5.278)	4.449 (4.069, 4.841)
Mean Squared Error	40.342 (35.192, 45.791)	33.119 (27.604, 39.339)
Relative Absolute Error	0.447 (0.399, 0.500)	0.404 (0.340, 0.471)
Correlation Coefficient	0.904(0.874, 0.927)	0.925 (0.894, 0.949)

CI: Confidence Interval.

**Table 4 cells-12-00032-t004:** Mean absolute error (MAE) from chronological age obtained in six different age groups divided into 10-year intervals.

Age Range (years)	N	Model without sjTREC	Model with sjTREC
age < 40	18	2.27	2.47
40 < age <= 50	18	4.17	3.89
50 < age <= 60	40	4.33	4.06
60 < age <= 70	44	5.05	4.60
70 < age <= 80	42	4.46	3.79
age > 80	32	8.21	6.90
All subjects	194	4.95	4.43

## Data Availability

The data that support the findings of this study are available from the corresponding author upon reasonable request.

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
