# Peer review of "A Blood-Based Molecular Clock for Biological Age Estimation"

_cells, 2022, doi:10.3390/cells12010032_

Round 1
Author Response
Thank you very much for your comments and suggestions: we will consider all of them while revising our manuscript. Here, we provide some feedbacks about your points. Your comments are in italic for readability. The modifications of text and tables were carried out using “Track Changes” option in Microsoft Word. We hope that revised manuscript could be suitable for publication.
Reviewer 1
The main question addressed by the research entitled “A blood-based molecular clock for biological age estimation’’ is a research article investigating biomarkers of biological age by quantification of DNA methylation levels of ELOVL2 and the amounts of sjTREC. There are several major concerns that should be addressed, showing as following:
- The logic behind to choose ELOVL2 as one of the two biomarkers? Though the authors briefly stated “it is included in most of current age prediction models and does not show tissue-specificity” (Page 2 Line 48-50), it is not strong enough. The authors may give more details why this specific gene is chosen and how its methylation status (the promoter region or the gene body) is correlated with ageing.
ELOVL2 encodes a transmembrane protein involved in the synthesis of very long polyunsaturated fatty acids (VLC-PUFAs). Several previous reports demonstrated that ELOVL fatty acid elongase 2 (Elovl2) is a gene whose epigenetic alterations are most highly correlated with age prediction (PMID: 23177740, PMID: 23061750, PMID: 31222117PMID: 25450787, PMID: 32973211). Within the top ten markers predictive of human epigenetic age, four are localized in the CpG islands in the regulatory element of the ELOVL2 gene, accounting for over 70% of the one “methylation clock” model developed by Hannum and co-workers [PMID: 29848354]. The age-dependent increase in Elovl2 regulatory region methylation is associated with concomitant downregulation of Elovl2 expression on mRNA and protein levels (PMID: 33043173). This age-related changes of Elovl2 methylation and gene expression were observed occur in multiple tissues in mouse, similarly to what was observed previously in humans (PMID: 29848354). In particular, in vitro studies suggest that the inhibition of ELOVL2 expression by RNA interference, was also associated with an increased senescence as well as decreased proliferation, both markers for aging. As expected, demethylation of ELOVL2 regulatory region was associated with an upregulatation of ELOVL2 expression that in turn was accompanied by the decreased senescence in a commonly used human fibroblast cell line. This suggests that the manipulation of ELOVL2 gene expression can have effects on aging in vitro. To validate this finding the same authors also demonstrated that a mutation able to eliminate the substrate specificity of Elovl2 elongation (Elovl2C234W) PMID: 23873268) was also able to accelerates aging in the mouse retina. Also in this case, it has been demonstrated that upregulation of Elovl2 expression may slow the functional aging. Finally, a recent work suggests that epigenetic variability of ELOVL2 regulatory region contributes to aging by regulating lipid metabolism (PMID: 35610223).
To better clarify all these aspects the “Introduction” section of the revised version of the manuscript has been modified and now reads [lines 51-57]:
“Several previous reports demonstrated that epigenetic variability of ELOVL2 regulatory region is highly correlated with age prediction (PMID: 23177740, PMID: 23061750, PMID: 31222117PMID: 25450787, PMID: 32973211). In fact, within the top ten markers predictive of human epigenetic age, four are localized in the CpG islands in the regulatory element of the ELOVL2 gene, accounting for over 70% of the one of the most validated epigenetic clocks so far developed (PMID: 29848354). For a functional point of view, in vitro and in vivo studies demonstrated that epigenetic variability of ELOVL2 promoter contributes to the variability of the aging process by regulating lipid metabolism (23873268, PMID: 35610223).”
- Since there is a study (reference [29]) using both ELOVL2 and sjTRECs to build a molecular clock for biological age prediction (Page 2 Line 64-65), what is the novelty of presented work comparing to [29] and other works?
The study carried out by Cho and colleagues analysed the variability of five epigenetic markers (ELOVL2, C1orf132, TRIM59, KLF14, and FHL2) together with the blood levels of signal-joint T-cell receptor excision circles (sjTRECs) in a small sample (100 individuals) form Korean population. For the first time we developed a more robust mathematical model (both form a statistical point of view and on the basis of the sample size) that exploiting only two of these biomarkers showed similar classification performances. With respect to the model proposed by Cho et al. or others previously developed models, our method presents several advantages: (i) require less amount of blood as usually occurs in forensic practice; (ii) it is less expansive; (iii) the underlying model is less complex.
- The authors claimed to build “an easy, cost-effective and reliable model” to predict biological age in the Abstract, how does the authors make the whole quantification process “easy and cost-effective” comparing to other works?
The referee is right. We stated that our model is easy, cost-effective and reliable since it is based on the analysis of only two well validated markers whose detection can be carried out using RT-PCR using commercial kits already available for both markers. Methylation analysis in particular is usually carried out using different DNA methylation technologies, including EpiTYPER® and pyrosequencing. The high cost associated with both these systems (equipment and reagents) is a constraining factor for the most part of forensic laboratories that require a more cost-effective technology. A cost-effective approach might also allow a re-analysis of the same sample, and the increase of the number of technical replicates (rarely performed in the published studies on this topic) has been demonstrated clearly improve the detection of the ELOVL2 methylation levels and, consequently, the corresponding age prediction models (PMID: 32973211).
To better clarify all these aspects the “Discussion” section of the revised version of the manuscript has been modified and now reads [lines 266-273]:
…”In fact, both markers can be analysed using several available RT-PCR methods (PMID: 36108573, PMID: 33281149, PMID: 35131731, PMID: 29574278, PMID: 21107596). In addition, technology such as single base extension can be easily incorporated into well-established capillary electrophoresis systems to analyse methylation levels of ELOVL2 gene (PMID: 30300865, PMID: 32325350, PMID: 35111197). As cost-effective approach, it might also allow a re-analysis of the same samples, and the increase of the number of technical replicates (rarely performed in the published studies on this topic) has been demonstrated clearly improve the detection of both markers and, consequently, the corresponding age prediction models (PMID: 32973211).”
- Can the authors give more details on the statistical analysis? The two models, one includes the sjTREC, the other excludes the sjTREC, were not given in the main text.
We thank the reviewer for this observation. We have revised the way we present the models in the “Development of a blood-based molecular clock” subsection. Particularly, we now provide direct links to the JADBio platform where the details of the two best models can be perused [lines 171-181].
- Is there any other parameters could affect the prediction accuracy? Such as when to draw the blood sample, the number and volume of blood donation or blood transplantation events for a specific individual, bisulfite sequencing methods, etc.
As mentioned before, the increase of the number of technical replicates (rarely performed in the published studies on this topic) has been demonstrated clearly improve the detection of the ELOVL2 methylation levels and, consequently, the corresponding age prediction models (PMID: 32973211). In fact, among the experimental factors affecting the robustness of DNA methylation analysis, the template amount in the PCR is one of the most important parameters. Storage, DNA input in the bisulfite conversion and type of bisulfite kit have low impact on methylation detection (PMID:27671843). As it regards the number and volume of blood donation or blood transplantation events, since we analysed DNA form buffy coats obtained more than ten years ago and frozen at − 80 °C before DNA extraction, we were not able to verify how such parameters could affect the prediction accuracy.
- There are a few minor issues:
- the full name of sjTREC (Page 1 Line 16).
- Is there an extra space between “[20].” and “Several”?(Page 2 Line 56)
- Is there an extra space between “whose” and “blood”?(Page 2 Line 71)
- Please specify the “Ethics Committee” in Page 2 Line 82.
- sJTREC in Page 3 Line 120.
- Please give the full name of MAE in the first time. (Page 6 Line 158)
- subsects or subjects, in Page 7 Line 189?
- exhibit or exhibiting, in Page 7 Line 191?
- Please re-edit the sentence in Page 7 Line 193-197.
- There should be a space between “[32].” and “(iii)” in Page 8 Line 207.
We thank the referee for all these suggestions
Reviewer 2 Report
In this short paper, the authors present a blood-based molecular clock for biological age estimation. They demonstrate that by including sjTREC levels in their model together with DNA methylation data, they can improve the age estimation, especially in older individuals.
This is an exciting and important topic that the authors address in a clear and relevant manner with a good summary of the field with relevant references. The results themselves are clearly presented, but the improvement of the model by including sjTREC levels seem rather modest. This is an important addition to the field and body of knowledge regarding molecular clocks, but I do not expect it to be very impactful.
The language holds a high quality and only requires minor edits. After some minor revisions, I recommend the manuscript for publication.
I have two specific comments and suggestions:
1. I am missing a simple table describing the demographic data of the studied population (for example this could be put next to chapter 2.1 where the samples are described). This information is presented in Fig 1 but not in a clear way. I do not think it is enough with mean age in this context. I would like to see the mean, standard deviation, median age and the range. And also separately presented for men and women (if at all relevant).
2. Related to above. The study population consists of 90 men and 104 women. How do these two groups compare to each other? If you use the age prediction model for men and women, separately, do you see any differences? I think this is a relevant question that I would like to see answered. Perhaps the reduced sample size will reduce the prediction quality, but this is a natural question when reading the manuscript.
Author Response
Thank you very much for your comments and suggestions: we will consider all of them while revising our manuscript. Here, we provide some feedbacks about your points. Your comments are in italic for readability. The modifications of text and tables were carried out using “Track Changes” option in Microsoft Word. We hope that revised manuscript could be suitable for publication.
In this short paper, the authors present a blood-based molecular clock for biological age estimation. They demonstrate that by including sjTREC levels in their model together with DNA methylation data, they can improve the age estimation, especially in older individuals.
This is an exciting and important topic that the authors address in a clear and relevant manner with a good summary of the field with relevant references. The results themselves are clearly presented, but the improvement of the model by including sjTREC levels seem rather modest. This is an important addition to the field and body of knowledge regarding molecular clocks, but I do not expect it to be very impactful.
The language holds a high quality and only requires minor edits. After some minor revisions, I recommend the manuscript for publication.
I have two specific comments and suggestions:
- I am missing a simple table describing the demographic data of the studied population (for example this could be put next to chapter 2.1 where the samples are described). This information is presented in Fig 1 but not in a clear way. I do not think it is enough with mean age in this context. I would like to see the mean, standard deviation, median age and the range. And also separately presented for men and women (if at all relevant).
We agree with the referee. The revised version of our Ms now includes an additional table (Table 1) that reports this crucial information.
- Related to above. The study population consists of 90 men and 104 women. How do these two groups compare to each other? If you use the age prediction model for men and women, separately, do you see any differences? I think this is a relevant question that I would like to see answered. Perhaps the reduced sample size will reduce the prediction quality, but this is a natural question when reading the manuscript.
We thank the reviewer for this interesting suggestion. We repeated the predictive analysis for the two genders separately [lines 186-191]. Indeed we observe a decrease in predictive performance, likely due to reduced sample size, however overall the results from these additional analyses are in agreement with our previous conclusion, with models using both sjTREC and ELOVL2 presenting a better mean absolute error than models using only ELOVL2.
Round 2
Reviewer 1 Report
The authors answered all my queries I am happy with the manuscript.